# Optimizing Plant Breeding Programs for Genomic Selection

Lance F. Merrick [1], Andrew W. Herr [1], Karansher S. Sandhu [1], Dennis N. Lozada [2] and Arron H. Carter [1,*]

1   Department of Crop and Soil Sciences, Washington State University, Pullman, WA 99164, USA;
    lance.merrick@wsu.edu (L.F.M.); andrew.herr@wsu.edu (A.W.H.); k.sandhu@wsu.edu (K.S.S.)
2   Department of Plant and Environmental Sciences, New Mexico State University, Las Cruces, NM 88003, USA;
    dlozada@nmsu.edu
*   Correspondence: ahcarter@wsu.edu; Tel.: +1-509-335-6198

**Abstract:** Plant geneticists and breeders have used marker technology since the 1980s in quantitative trait locus (QTL) identification. Marker-assisted selection is effective for large-effect QTL but has been challenging to use with quantitative traits controlled by multiple minor effect alleles. Therefore, genomic selection (GS) was proposed to estimate all markers simultaneously, thereby capturing all their effects. However, breeding programs are still struggling to identify the best strategy to implement it into their programs. Traditional breeding programs need to be optimized to implement GS effectively. This review explores the optimization of breeding programs for variety release based on aspects of the breeder's equation. Optimizations include reorganizing field designs, training populations, increasing the number of lines evaluated, and leveraging the large amount of genomic and phenotypic data collected across different growing seasons and environments to increase heritability estimates, selection intensity, and selection accuracy. Breeding programs can leverage their phenotypic and genotypic data to maximize genetic gain and selection accuracy through GS methods utilizing multi-trait and, multi-environment models, high-throughput phenotyping, and deep learning approaches. Overall, this review describes various methods that plant breeders can utilize to increase genetic gains and effectively implement GS in breeding.

**Keywords:** plant breeding; speed breeding; training population; field design; multi-environment; multi-trait; deep learning; high-throughput phenotyping; genetic gain

## 1. Genomic Selection

With the advent of marker technology in the 1980s, geneticists and breeders have used marker technology to improve selection strategy and efficiency in breeding programs [1]. Marker technologies were first used in quantitative trait loci (QTL) identification [2–4]. The identification of QTLs allowed marker-assisted selection (MAS) and introgression to select and deploy specific marker-linked traits in a population efficiently [5]. Marker use is effective for large-effect QTL but has proven to be challenging to use with quantitative traits that are controlled by multiple genes with minor effects. Previous methods to deal with quantitative traits were developed, such as a multi-marker MAS system, but it is difficult to identify and account for all the allele effects [6,7]. Therefore, Meuwissen et al. [8] proposed the idea of simultaneously estimating all markers regardless of "significance", and thereby, capturing all their effects. Meuwissen et al. [8] coined this method "Genomic Selection" (GS), which has also been referred to as genome-wide selection or genomic prediction [1,8].

The first successful studies using GS were in dairy cattle (*Bos taurus*) breeding, where it was implemented to market bulls [9]. Until recently, plant breeders have typically relied on phenotypic selection (PS). However, this trend has been changing within the last decade. The first GS study in plant breeding was conducted in maize (*Zea mays* L.) [10]; after which the approach has been successfully implemented in other cereal grains such as wheat (*Triticum aestivum* L.), barley (*Hordeum vulgare* L.), and oat (*Avena sativa* L.) [10–13].

## 2. Genetic Gain

Genetic gain, also known as the genetic response ($R$), is calculated by what is known as the breeder's equation, $R = \frac{ir\sigma_A}{t}$, where $i$ is the selection intensity; $\sigma_A$ is the square root of the additive genetic variance; $r$ is the selection accuracy, which is the equivalent to narrow-sense heritability ($h^2$) in PS; and $t$ is the cycle time [14,15]. Plant breeders use the breeder's equation to increase the genetic gain of their breeding program. By increasing one of the components in the numerator or decreasing cycle time ($t$), a breeder can increase genetic gain. The increase of genetic gain using PS is difficult for traits with low heritability [12]. Consequently, selection on traits with low heritability, such as grain yield, is completed at the later stages of a breeding program. If the environmental effect on a trait is high enough, such as drought, disease, or other adverse conditions, the selection based on PS will be challenging. The limitations of changing the denominator ($t$) are affected by the ability to evaluate the gene in question. An example of this limitation is grain yield, in which the trait can only be measured after the full maturity of the plant. One of the ways to maximize the genetic gain is to increase the selection accuracy in a breeding cycle, which can be accomplished by different molecular genetics approaches such as MAS or GS [5,8].

The traditional breeding program focuses on selecting varieties for release. Gaynor et al. [16] proposed reorganizing the traditional breeding program into two parts: the product development (PD) component which is similar to traditional breeding programs, and a population improvement component to utilize recurrent GS. Pipelines for PD have been extensively studied for the implementation of GS because it is easily integrated into existing structures of breeding programs [17–19]. Genomic selection allows the use of genomic-estimated breeding values (GEBVs) in lieu of phenotypic data. Replacing phenotypes with GEBVs allows the restructuring of breeding programs. Genomic selection can simply replace phenotypic or MAS for selection purposes [20,21]. However, this strategy does not necessarily increase genetic gain for certain traits, such as grain yield, due to the lack of increase in selection accuracy compared to PS. There are several opportunities to increase the genetic gain by optimizing breeding programs for GS. These include reorganizing field designs, increasing the number of lines evaluated, and leveraging the large amount of genomic and phenotypic data collected across different growing seasons and environments to increase heritability estimates, selection intensity, and selection accuracy [17–19]. The trait data consist of phenotypic values collected from multiple environments, multiple traits, and high-throughput phenotyping. Recent developments of multi-trait, multi-environment GS models are poised to leverage the large amount of phenotypic data in breeding programs to improve selection accuracy for quantitative traits [22,23]. In this review, we explore the optimization of breeding programs for GS for a wheat (inbred crop) breeding program based on the components of the breeder's equation.

## 3. Breeding Program Optimization

Wheat breeding PD programs focus on developing inbred lines for release as inbred varieties. Traditionally, after crossing and population improvement, inbred lines are developed either through self-pollination or doubled haploids. The inbred lines are then phenotyped in headrows and field trials before being selected as parents in the crossing block. This method takes up to four to six years in wheat, depending on the breeding program structure and preference of the breeder. In the Washington State University Winter Wheat breeding program, for example, the inbred lines are developed through both self-pollination and doubled haploids. Headrows are the first stage of phenotyping and happen in the fourth year, followed by unreplicated preliminary yield trials (PYT) in the fifth year. It takes until the sixth year to start with replicated field trials in the advanced trials. Inbred lines are in replicated yield trials up to five more years when varieties are released at the end of the 11th year. The long length of the breeding program allows for ample opportunity to optimize the breeding program (Figure 1).

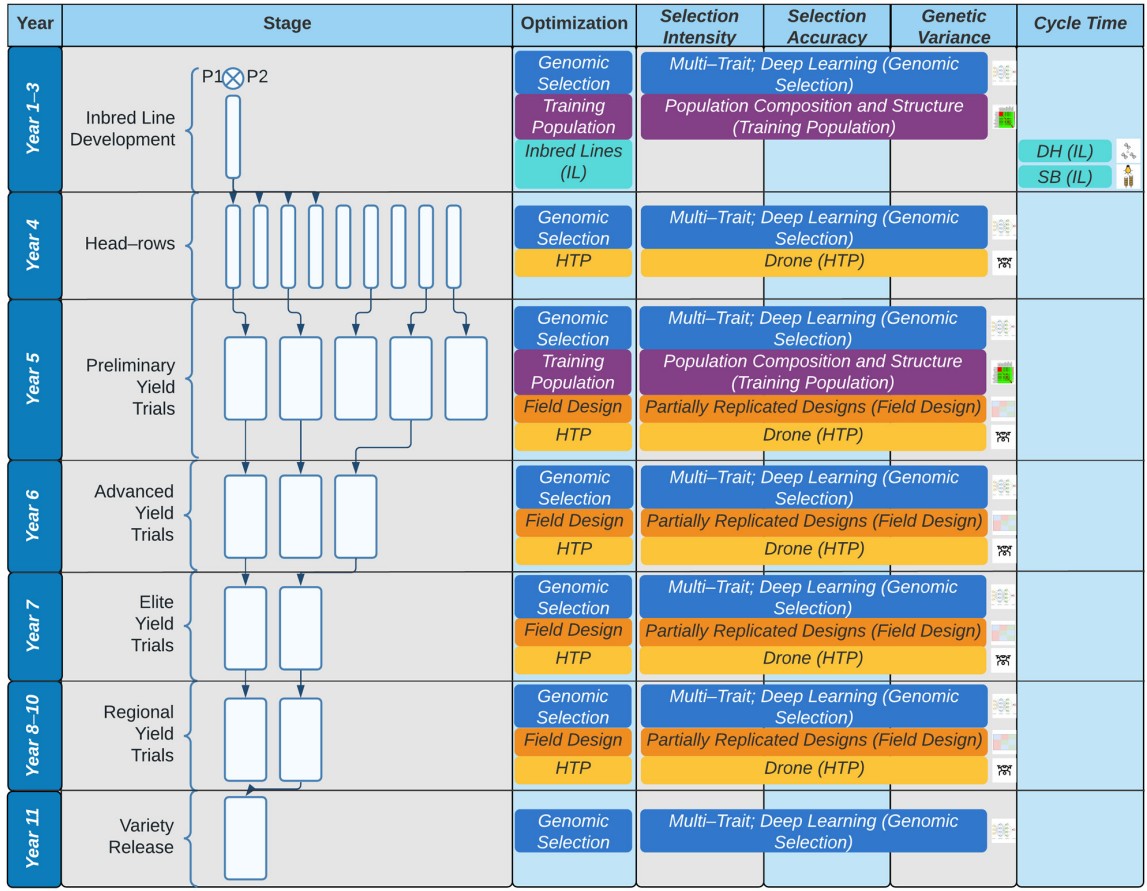

**Figure 1.** Optimization of the traditional breeding pipeline and product development based on an 11–year breeding program from parental crossing to variety release. The effect of each component of optimization (genomic selection, training population design, inbred line development, field design, high-throughput phenotyping (HTP) on different aspects of the breeder's equation (selection intensity, selection accuracy, genetic variance, and cycle time) is shown by the coverage of the method of optimization within the respective column of the different factors of the breeder's equation. For example, for Years 1–3 of the breeding cycle, the composition and structure of the training population (purple) affect both selection accuracy and genetic variance, whereas the choice of genomic selection models affects the intensity of selection, prediction accuracy, and genetic variance.

### 3.1. Speed Breeding and Doubled Haploids

The optimization process for the PD pipeline can start immediately after the hybridization of the parental lines. The first optimization is in inbred development, which can decrease the length of the PD pipeline, and therefore cycle length. This can be achieved through traditional self-pollination such as single-seed descent (SSD) or rapid fixation of lines via doubled haploids (DH). The creation of inbred lines allows the within-line variation to be minimized while increasing between line variation to allow maximum genetic gain via selection. However, one of the most recent developments in inbred development is speed breeding which does not require specialized labs for in vitro culturing and can be applied over diverse germplasm, unlike DH production [24]. Speed breeding accelerates generation advancement by manipulating growing conditions under prolonged photoperiod and through temperature control to increase the rate of development and growth in plants [25]. Therefore, speed breeding has the ability to reduce the generation time and accelerate breeding programs. Using speed breeding, Watson et al. [25] were able to reach six generations per year in wheat. After the creation of the inbred lines, they can be implemented into the training population and phenotyped in field trials. In another study, Watson et al. [26] integrated multivariate GS and speed breeding to reduce the breeding

cycle by quickly producing inbred lines and integrate indirect selection for traits such as height and flowering time as well as yield-related traits before field trials.

### 3.2. Training Population Design

The predictive ability of GS models is primarily dependent on the training population used for predictions. Optimizing training populations influence the genetic variance and selection accuracy factors in the breeder's equation. The training population should be developed once the test (validation) population and goal of prediction are identified. Training populations range from biparental populations in some of the earlier GS studies [8,27,28]; to using exotic or diverse populations [29] or using the breeding lines themselves [30–32]. Regardless of the goal, the training population needs to balance the costs of phenotyping while maximizing predictive ability [33].

The composition and structure of the training population directly relate to prediction accuracy [34,35]. Utilizing GS within bi-parental populations reduces the number of lines phenotyped and genotyped for high levels of accuracy due to high levels of linkage disequilibrium [36]. They can also be readily applied to recurrent selection to predict future cycles of selection from intermating related lines [12,13,27,37]. An advantage of biparental populations is the high level of genetic relatedness between the training and test populations. However, when combing unrelated lines from various pedigrees, the prediction accuracy generally decreases [38–40]. Training populations based on bi-parental populations and selecting within individual families have limited applicability within a large breeding program, especially when resources are limited [17]. There have been many comparisons of training populations within breeding programs [30,32,41,42].

Another method is to use diversity panels that are commonly used and developed for genome-wide association studies [31,32,41,42]. These populations generally have a large population structure and reduced prediction accuracy [32]. When there is greater diversity within a training population, more lines and markers are needed to increase accuracy. This can be difficult to do within traditional breeding programs, as it requires the cultivation of an additional set of lines just for the training population. An alternative is to use the breeding program itself as the training population [30]. As lines being tested within the breeding program are genotyped and phenotyped, they begin to develop the basis for the training population, and over time, large datasets are collected with little additional work. However, with most breeding population trials, many lines share some degree of relatedness, which can increase GS prediction accuracy. Ensuring lines that are highly related with a limited population structure is ideal in training population optimization [35].

One of the most important factors in determining the accuracy of GS is the population size [34–36]. Training population size affects genetic variance, and large diversity and genetic variance require larger training populations [35,43]. The more diverse a training population is, the larger the number of genotypes needed to account for the large genetic diversity, specifically for low heritability traits [44]. Training population size impacts accuracy more than marker number or density. The size of the training population and marker density is dependent on the QTL number and heritability of the trait. Low heritability traits require larger population sizes, but results have shown that there are effective population sizes for even extremely low heritability traits, and these traits can still be accurately predicted [36,45].

Specifically, in a breeding program, the goal for optimum training population size is to create a population that maximizes prediction accuracy with the least number of individuals possible. Finding the optimum size reduces the cost of genotyping and phenotyping, and hence increases the efficiency of plant breeding programs. It has been shown that prediction accuracy improves as the number of genotypes increases due to the reduction of bias and variance of marker effect estimates [46,47]. Smaller training sets have the risk of overestimating the genotypic effect when predicting larger validation sets. In general, prediction accuracy in wheat has been shown to increase with the increase of training population size, with the highest accuracy around 300 genotypes and a gradual plateau

after 300 lines [35,48]. Muleta et al. [29] further demonstrated an increase in accuracy as the training population increased in both elite breeding lines and diversity panels.

Methods to identify optimized training populations within populations have been compared. For example, Tiede and Smith [13] compared a stratified sampling method, Gmean, CDmean, and selection of training populations by genetic algorithms (STPGA) for predicting yield and disease resistance in barley. Stratified sampling creates training populations based on clusters from population structures already existing in the breeding program. Gmean calculates the means of the training population and validation population within a genomic-relationship matrix, and lines within the training population with the highest mean relationships to the validation population were used. The CDmean maximizes the coefficient of determination, whereas STPGA utilizes genotypic and phenotypic data to minimize prediction error variance among selection candidates. While Gmean was found to perform best for grain yield, STPGA performed best for deoxynivalenol. The best optimization is dependent on the population, and the breeder needs to compare the methods to determine the best fit for the trait in question [13]. In another study, a weighted relationship matrix with stratified sampling was shown to be the best method for training population optimization, especially in forward predictions of distant generations [49].

Within breeding programs, large amounts of phenotyped lines are not usually a constraint. In crops such as wheat, pooling together many small families for the training population is advised, whereas for hybrid crops such as maize, choosing a few families with a large number of lines is more appropriate [33]. The number of lines per family that typically reach the field trials is small, which reduces the ability to form a training population based on a few families with many lines. Therefore, by combining breeding populations trials with various pedigrees and genetic relatedness, many lines can form a training population, especially if their ancestral pedigrees have been genotyped. This can help optimize genetic relatedness and training population selection through genomic relationship (GRM) or marker matrices [33]. By leveraging and optimizing the breeding program for GS purposes, large training populations with shared ancestry can be developed. However, in order to do so, one needs to design the program and models to deal with combining trials and environments.

*3.3. Field Design*

Once lines are considered fixed in terms of allele frequency, the PD and selection can begin on a large scale. The inbred lines still need to be phenotyped in headrows and field trials, and GS can be fully implemented for selection. There have been various developments in optimizing PD pipelines for GS. Breeding programs have limited resources to allocate and can limit the size, number of replications, and locations of field trials. Ultimately, these factors influence the ability to estimate marker effects and genetic gain. In general selection terms, screening more lines increases the chance to identify high-performing lines while replicating individual lines creates more accurate genotypic estimates [33]. Due to increased genotypic estimates, the phenotypic variation is decreased, which increases heritability. However, the increase in heritability plateaus and further optimization of field designs need to be completed [50].

Individual trial designs are just as important to increase heritability and phenotypic data quality. Field trials can be optimized to increase heritability, genetic variance, selection accuracy, and an increase in lines screened which can improve selection intensity and genetic gain. This is due to GS model accuracy being contingent on the quality of the phenotypic data and control of spatial variation to increase heritability and selection accuracy [19,51].

Spatial variation can be accounted for in the initial design of the trial through blocking, checks, and analysis via spatial correction. Blocking and randomization designs range from the basic with no blocking, to completely randomized design and randomized complete block through the row-column designs. Incomplete block designs are a popular method to increase the number of lines screened with limited resources, such as the alpha-lattice

design [52]. In early generations, un-replicated or augmented complete block design or incomplete block designs hinge on un-replicated genotypes with replicated checks [53].

Recently, other augmented designs have been explored, such as the partially replicated experimental design (PREP) that uses replicated genotypes instead of checks which help avoid bias [54]. The PREP design was shown to display optimum accuracies using a fixed budget and resources compared to other designs [54]. Moreover, the PREP design spreads replicate across locations instead of replicating all lines in all locations and thus increases the number of lines phenotyped. The augmented PREP (APREP) design extended the original PREP design with multi-environments in which lines are replicated in a single environment, and all other environments are un-replicated [55].

Recent simulations showed that completely replicated designs such as the alpha-lattice and row-column designs increased GS prediction accuracy over all other partially replicated or un-replicated designs [19]. This was due to the increase in heritability from replication and the ability to partition genotypic and environmental effects and reduce error. Overall, the alpha-lattice design performed the best over all heritability and genotype-by-environment (GE) scenarios for GS prediction accuracy. The PREP designs outperformed the un-replicated designs for GS prediction accuracy and had the highest response to selection when the heritability was low, and the population size was large [19].

Spatial variation can also be accounted for via spatial correction. Spatial corrections during statistical analysis range from the common two-dimensional autoregressive model for spatial variation for a row-column design [56] and a two-dimensional spline model [51,57]. Additionally, spatial correction can also be analyzed with nearest neighbor analysis and the one-dimensional linear variance model plus the incomplete block model, additive or separable form [55,58,59]. In Ward et al. [60] both one-dimensional and two-dimensional autoregressive models led to large increases in heritability, but only a small to non-significant increase in GS prediction accuracy. Additionally, Hoefler et al. [19] noted that spatial corrections including the two-dimensional autoregressive model had a minimal increase in GS prediction accuracy when implemented to the range of field designs stated previously but would be most beneficial in large trials. Therefore, the advantage of spatial corrections is case-specific [51,55,57,61,62].

## 4. Leveraging Phenotypic Data

### 4.1. Multi-Environment Models

Genomic selection has been shown to be accurate in single environments, but most prediction models do not have the predictive power to make selections across multiple environments or account for genotype-by-environment (GE) interaction. In plant breeding, GE plays a major role in the variation of certain traits, such as grain yield. Phenotypic variation can be divided into genetic and non-genetic effects [63]. GE increases phenotypic variation without increasing genetic variation and thus, decreasing heritability [64]. Adequate experimental designs and phenotypic adjustments historically accounted for non-genetic effects. However, accounting for GE in prediction models, such as using genomic best linear unbiased prediction (GBLUP), is important to optimize a breeding program to account for the combination of trials over multiple years and locations, which can ultimately increase selection accuracy and genetic variance (Table 1).

One of the simplest methods to account for GE in GS models is the two-step adjustments (Two-step GBLUP). In this process, field and environmental corrections are applied to the phenotypic data before integrating them into the GS models. Predictions using two-step models were shown to be equivalent in prediction accuracy to single-step GBLUP models that integrate covariates or GE marker interaction into the GS model [60]. In another method to optimize environments for GS purposes, Lado et al. [69] grouped environments based on genotype-by-GE biplots to create mega-environments and optimized variance-covariance matrices across environments (GGE GBLUP) with low GE to better predict genotype performance in untested environments.

**Table 1.** Genomic selection (GS) models leveraging phenotypic data for multi-trait (MT), genotype-by-environment interaction (GE), and multi-trait, multi-environment (MTME) models that have shown an increase in prediction accuracy over single-environment, single-trait, GS models.

| Model | Factor | Description | Software Package (Programming Language) | Reference(s) |
|---|---|---|---|---|
| Two-Step Genomic best linear unbiased prediction (GBLUP) | GE | Environmental and Phenotypic Adjustments made prior to GS using a linear mixed model. | ASREML (R) BGLR (R) | [60] |
| Single-Step GBLUP | GE, MT, MTME | GE GBLUP models using compound symmetry, heterogeneous variance, and factor-analytic unstructured models. | ASREML (R) BGLR (R) | [60] |
| Factor-Analytic (FA) GBLUP | GE | FA GE GBLUP Model | ASREML (R) BGLR (R) | [65] |
| Crop-Growth (CG) covariate GBLUP | GE | CG model derived stress environmental covariates (EC) using the Kronecker product | ASREML (R) BGLR (R) | [66] |
| Reaction-Norm (RN) GBLUP | GE | RN model where the main and interaction effects of markers and environmental covariates are introduced using highly dimensional random variance-covariance structures | BGLR (R) | [67] |
| RN model for phenotypic plasticity (PP) GBLUP | GE | RN GBLUP model for phenotypic plasticity | rrBLUP (R) BGLR (R) | [68] |
| Enviromic-aided (ET) GBLUP | GE | EC GBLUP using Envirotyping | EnvRtype (R) | [63] |
| Genotype-by-Genotype-Environment (GGE) GBLUP | GE | GE based on GGE Mega-environments and additive main-effects and multiplicative interaction (AMMI) using the Kronecker product | rrBLUP (R) BGLR (R) | [69] |
| Marker-Environment Interaction (ME) GBLUP) | GE | ME model that decomposes the marker effects into components common across environments and environment-specific deviations. | BGLR (R) | [70] |
| ME Linear Genome-Based Kernel (GB) GBLUP | GE | ME with the Linear GB kernel | BGLR (R) | [71] |
| ME Gaussian Kernel (GK) GBLUP | GE | ME with the Gaussian GK kernel | BGLR (R) | [71] |

**Table 1.** *Cont.*

| Model | Factor | Description | Software Package (Programming Language) | Reference(s) |
|---|---|---|---|---|
| GB GBLUP | GE, MT, MTME | GE using Kronecker product with the Linear GB GBLUP model | BGLR (R); BMTME (R) | [22,64,72,73] |
| GK GBLUP | GE, MT, MTME | GE using Kronecker product with the Gaussian GK GBLUP model | BGLR (R); BMTME (R) | [22,64,72,73] |
| BGGE GB GBLUP | GE | GE using Hadamard product with the Linear GB GBLUP model | BGGE (R) | [74] |
| BGGE GK GBLUP | GE | GE using Hadamard product with the Gaussian GK GBLUP model | BGGE (R) | [74] |
| Approximate Kernel (AK) RN GBLUP | GE | Sparse Approximate Model using the RN GBLUP model | BGLR (R) | [75] |
| AK GBLUP | GE, MT, MTME | Sparse Approximate Model using the Kronecker product for GB and GK GBLUP along with various other kernels | BGLR (R) | [76] |
| Multi-Layer Perceptron (MLP) | GE, MT, MTME | Deep learning MLP that uses a combination of input, hidden, and output layers using a large number of neurons for building the relationship between the predictors and output that has the ability to incorporate GB and other kernels and use any GE method. | TensorFlow (R and Python) and Keras (R and Python) | [72,73] |

Further, several models implicitly account for the GE effect within the GS model itself. One of the first methods to deal with GE was implementing factor analytic (FA GBLUP) models that are flexible for the genetic variance-covariance for environments [65]. Jarquín et al. [67] extended the GE GBLUP model by using genetic markers and environmental covariates (EC) to increase prediction accuracy significantly. Further, GS has been modeled using reaction-norm (RN) from ECs, which is a linearized response from genotypes for a target environmental gradient and can be modeled explicitly as genotype-specific covariates using factorial regressions (RN GBLUP) [66,77,78]. Another approach for utilizing RN models is to model phenotypic plasticity (PP GBLUP) [68]. In understanding phenotypic plasticity, they identified environmental indices to connect environments quantitatively using GS with RN parameters. Environmental covariates can also be modeled using crop growth models (CG GBLUP) and deep kernel approaches [66,79–81]. Recently, environmental covariates using geographic information system information have been used to better deal with GE [82]. Further, Costa-Neto et al. [63] developed an R package called "EnvRtype" to integrate large-scale envirotyping (enviromics) into quantitative genomics for implementation in GS. EnvRtyping was utilized for enviromic-aided GS (ET GBLUP) and outperformed conventional GBLUP for predicting grain yield in maize into untested environments.

The next development for multi-environment GS models was accounting for the effects of Marker-Environment interaction (ME). GE can be modeled explicitly by modeling

interactions between markers and environments using ME in GBLUP [70]. The ME model decomposes effects into components common across environments and environment-specific deviations. This can be used to model stable effects across environments and environment-specific interactions [70]. Additionally, ME can be modeled using GBLUP (ME GBLUP) or variable selection methods. However, the ME approach has limitations on covariance patterns, making the model positive and homogenous and best suited for joint analysis of positively correlated environments.

Additionally, GE can be modeled using genome-based kernels, pedigree, and GRMs. In genomic prediction, linear models incorporate genetic values as linear combinations of markers so that the linear genome-based (GB) kernel is equivalent to ME. However, departures from linearity happen regularly in GS due to complex interactions among genes and their interaction with the environments and can be addressed using nonlinear kernels such as Gaussian kernels (GK). Cuevas et al. [71] compared methods that applied the ME GBLUP method of Lopez-Cruz et al. [70] using the linear GB (ME GB GBLUP) and the nonlinear GK (ME GK GBLUP) model and displayed an increase in accuracy by up to 17% over ME GBLUP. However, the ME GK and GB GBLUP models also assume positively correlated environments [71]. These models assume a positive correlation because they use the Hadamard product for modeling GE and exchange information between environments using the variance-covariance matrix of the main effects. This method has an advantage when the number of lines in each environment is the same but can also be extended to an unbalanced number of lines in each environment, as shown in Bandeira e Sousa et al. [83].

In contrast, GE GBLUP can be accomplished by using the Kronecker product of the variance-covariance matrices of the relationships between environments and GRMs. The Kronecker method allows negative correlations between environments. Bayesian regression models for GE previously used the Kronecker method for unstructured variance-covariance matrices between environments and genomic kernels using both the GK (GK GBLUP) and GB (GB GBLUP) kernels [84]. However, the Bayesian models used to implement the kernels increased computing time. To overcome this, Granato et al. [74] created the Bayesian Genomic GE ("BGGE") package in R to fit Bayesian models with homogenous error variances proposed in Jarquín et al. [67] and Lopez-Cruz et al. [70]. Cuevas et al. [85] compared the Hadamard product ME GBLUP model with the ME GB GBLUP and ME GK GBLUP kernels implemented in BGGE to the GB GBLUP and GK GBLUP kernels using the Kronecker product method. The Hadamard product models decreased computing time but proved the advantages of the Kronecker product models for environments with zero to negative correlations while confirming the increase in accuracy of using the GK over the GB [85].

Another useful evolution in modeling GE is using sparse matrices to create approximate kernels to reduce computational time with comparable prediction accuracy (AK GBLUP) [75]. Approximate kernels are advantageous for large datasets requiring intense computation and matrix decomposition time [75]. The prediction accuracy of the approximate kernels depends on the number of subset lines and the decrease in eigenvalue decomposition of the GRM. Further, Montesinos-López et al. [86] outlined the implementation of sparse matrices from Cuevas et al. [75]. They integrated them with the Bayesian methods from Cuevas et al. [84] to create linear, polynomial, sigmoid, Gaussian, and Arc-cosines with one or more hidden layers and exponential kernels in both a multi-environment and multi-trait framework.

*4.2. Multi-Trait Models*

In addition to GE, breeders simultaneously select multiple traits to advance lines. Genomic selection has been mainly used for the prediction of single traits, but the ability to select for multiple traits would be advantageous when trying to evaluate and select genotypes based on combinations of yield components, end-use quality, or disease traits. Additionally, multiple traits may be positively or negatively correlated, which increases the complexity of improving multiple traits simultaneously [22]. The joint analysis of multiple

traits takes advantage of the genetic correlation between the traits, which can increase prediction accuracy, specifically for lowly heritable traits that are genetically correlated with highly heritable traits, and ultimately increases selection accuracy and genetic variance similar to accounting for GE [22,87–89].

Multi-trait (MT) analysis can also facilitate predicting untested lines and unobserved traits. MT or multivariate models can take advantage of correlation to increase accuracy, statistical power, parameter estimation, and ultimately reduce selection bias when implementing MT selection [22]. MT models range from GBLUP, Bayesian, and, recently, deep learning (DL) models [22,23,72]. The MT models have been used in BLUP models [90]. According to Calus and Veerkamp [91], multivariate Bayes Stochastic Search Variable Selection outperformed Bayes Cπ and GBLUP, when the trait had major QTLs and the MT models had higher prediction accuracies compared to single-trait models. However, for polygenic traits, the multivariate trait models performed similarly to the single univariate models [87]. Multivariate models predict better when the traits in question are genetically correlated with each other [87]. Accuracy of GS for low heritability traits (e.g., grain yield) can also be significantly increased by multivariate models when a correlated highly heritability trait is available [87,91]. MT models can improve indirect selection due to increased genetic correlation estimates [22,72,92]. Montesinos-López et al. [76] showed the higher the genetic correlation between traits, the higher the prediction accuracy and benefit of MT over single trait models.

Using GS to predict selection indices is another way to select multiple traits. Index selection involves selecting multiple traits simultaneously based on a selection index [93]. A selection index integrates and weights multiple traits to create greater genetic gain as compared to independent trait selection. Selection indices can use marker sets as indirect selection traits. Using MAS and linear stepwise regression models violates selection index assumptions of multivariate normality since selection is based on only a few large-effect loci. However, GS does not violate this assumption since it simultaneously predicts all marker effects [94]. Genotypic selection indices have been shown to be more efficient than PS indices in both simulated and empirical data [95].

### 4.3. Multi-Environment, Multi-Trait Approaches

The advantages of GE and MT models can be combined into multi-trait, multi-environment (MTME) models. In Ward et al. [60], a single-step MTME GBLUP model using unstructured variance-covariance matrices between residuals, main effects, and GE implemented in ASReml displayed an increase in accuracy for lowly heritability traits. The common MT and GE GBLUP models are unable to estimate separable unstructured variance-covariance matrices for a three-way interaction term. The multivariate GBLUP model has to assume one of the variance-covariance matrices as a new variable created by merging two of the three factors and estimating the covariance matrix with two components that cannot be separated [22]. The MT and GE Bayesian models have been extensively used, as discussed previously, but a Bayesian MTME model (BMTME) was not developed until Montesinos-López et al. [22] unified the two models. The BMTME model can be advantageous when individuals are phenotyped for all traits in one environment but not in the others, and vice-versa for environments. The BMTME models were evaluated using grain yield, disease index, and plant height using multiple covariance structures. Montesinos-López et al. [76] found that when trait correlations are above 0.50, the unstructured covariance matrix outperformed the diagonal and standard covariance matrices. The standard covariance matrices are performed similarly to the other covariance structures when the correlation is low. The MTME models allow GS models to take advantage of common breeding program scenarios when lines are phenotyped for multiple traits in multiple environments and allow the leverage of compiling all available data to predict using a single model. As an approach, MTME models demonstrated an improved accuracy over single trait models for a variety of agronomic traits, including grain yield and enhanced resource efficiency in wheat [96].

### *4.4. Deep Learning*

The latest development in GS is the implementation of machine and deep learning models, which can significantly impact multi-environment and multi-trait applications to increase genetic variance and selection accuracy. Machine learning (ML) models use statistical methods to learn iteratively for improved performance and accuracy without explicitly being told what to do [72]. Deep learning is a form of ML that uses densely connected artificial neural networks with multiple layers linked using subsets of non-linear semi-parametric models [97]. Neural networks were modeled after the complexities and biological networks of brain neurons [98]. The "deep" in DL, refers to the use of multiple combinations of layers that transform data such as marker information [72]. These models can be used for both classification and regression and have shown to be comparable or even increase prediction accuracy to linear regression GS models [23,72,99–101]. One of the most commonly used DL models is a multi-layer perceptron (MLP) neural network model, also referred to as a feed-forward neural network [101]. The MLP uses densely connected layers, also called networks, composed of input, output, and multiple hidden layers. Weighted units or neurons are then connected in a network with nonlinear activation functions that are able to accurately predict the genetic architecture of a trait [101]. In theory, MLP may require a large number of hidden layers especially when the data is nonlinear [102]. However, in practice, one layer with many neurons is enough to approximate the desired degree of accuracy with only two layers to better capture the non-linear interactions [102]. Another common DL model, convolutional neural networks (CNN), is a special case of DL models where hidden layers consist of convolutional layers that are flattened and fully connected via dense layers. The CNNs were first proposed in GS to account for inputs that are associated, such as LD between markers [101]. In another study, MLP outperformed CNN and rrBLUP models in a spring wheat nested association mapping population across five different agronomic traits [101].

In the context of GE, Montesinos-López et al. [72] compared GBLUP and DL MLP GE models. When GE was accounted for, the GBLUP model obtained the highest accuracy in eight of the nine data sets, but the DL MLP had the highest accuracy in six out of the nine data sets when GE was ignored. The increase in accuracy in the DL MLP when GE was not implicitly modeled was accounted for by the ability of DL MLP models to capture complex relationships in the data without explicitly accounting for them. The lack of improvement in accuracy for the DL was due to the scarcity of data in the smaller data sets when using grid-search and hyperparameter optimization. Further disadvantages of the DL were due to the increased computation time to optimize the models, increase in the number of layers and units, and a demand for higher experience to implement them. Additionally, Montesinos-López et al. [73] compared an MTME DL MLP model to the MTME GBLUP models outlined in Montesinos-López et al. [22] using the BMTME package with comparisons to both models with and without the inclusion of GE. Montesinos-López et al. [73] showed similar results to Montesinos-López et al. [72] and found that MTME GBLUP models displayed higher accuracy in two of three data sets. Further, the MTME GBLUP displayed higher accuracy across environments with GE, and the MTME MLP displayed higher accuracy across environments without the inclusion of GE. However, in contrast to the single trait GE comparison in Montesinos-López et al. [72], the MTME MLP required less computational resources than the MTME GBLUP models. Therefore, with contrasting results, the DL models are currently an addition to the GS toolbox rather than a replacement, and models for GE should be compared and used on a case-by-case basis.

### *4.5. High-Throughput Phenotyping*

High-throughput phenotyping spectral data can also be integrated into univariate and MT models and can significantly impact genetic variance, selection intensity, and selection accuracy. Spectral reflectance indices (SRI) are standardized secondary traits highly associated with primary traits of interest and difficult and expensive traits to phenotype. These secondary traits usually have higher genetic and phenotypic correlation, high

heritability values, and are easier to phenotype than complex lowly heritable traits. This association and the underlying prediction improvement in MT models lie in the genetic correlation of SRIs and grain yield [23,87,103]. Rutkoski et al. [104] were among the first to model SRIs collected using HTP tools using a multivariate GBLUP model and showed an increase in prediction accuracy for grain yield by 70% (Table 2). SRIs allow for the simple addition of a single or a few secondary traits in MT models. However, another common form of HTP data is hyperspectral. Hyperspectral sensors capture thousands of points across the reflectance spectrum. Relationship matrixes generated from hyperspectral data have been shown to model both genetic main effects and GE interaction effects [105]. By incorporating marker, pedigree, and HTP data, an additive benefit was found when evaluating genetic and GE effects across a breeding program. This, in turn, increased prediction ability for grain yield when multiple kernels were used in GBLUP models. Furthermore, Sandhu et al. [23] demonstrated that MT ML random forest and DL MLP models increased prediction accuracy for grain yield and grain protein content by up to 29 and 15%, respectively, when SRIs were incorporated into the model.

**Table 2.** Univariate and multivariate genomic selection (GS) models that have been used to incorporate high-throughput phenotyping (HTP) spectral reflectance indices (SRI).

| Model | Description | Software Package (Programming Language) | Reference(s) |
|---|---|---|---|
| Genomic best linear unbiased prediction (GBLUP) | The GBLUP model that uses GRMs for predicting their performance. In addition, has the ability to use single and multi-kernel models combining hyperspectral and genomic marker information | ASReml (R); BGLR (R) | [23,103–105] |
| Bayesian | Bayesian models (Bayes A, Bayes B, Bayes C, Bayes Cπ, Bayes D, Bayes Lasso, Bayes Ridge Regression) that use marker effects by assuming a scaled inverted chi-square distribution, scaled t distribution, or double exponential distribution for variance parameters to model marker effects. | BLGR(R); BMTME (R) | [23] |
| Elastic Net (EN) | EN is the intermediate between ridge regression and lasso using an average weight penalty for marker effect estimations. | glmnet (R) | [103] |
| Partial least square regression (PLSR) | PLSR is a dimensional reduction approach that uses latent variables derived from predictors to link with the response variables. | pls (R) | [103] |
| Random Forest (RF) | RF uses a network of the tree with varying number of nodes, resampling, and depth for building the final tree regression for predictions | caret (R); Scikit-learn (Python) | [23] |

**Table 2.** *Cont.*

| Model | Description | Software Package (Programming Language) | Reference(s) |
|---|---|---|---|
| Support-Vector Machine (SVM) | SVM is a non-parametric method that uses kernels functions, and cost functions to model hyperplanes for predictions. | caret (R); Scikit-learn (Python) | [23] |
| Convolutional Neural Network (CNN) | Deep learning CNN that uses convolutional, flattening, pooling, and dense layers for predicting using kernels to reduce the excess predictors from the model. | caret (R); Keras (R and Python); Scikit-learn (Python) | [23] |
| Multi-layer Perceptron (MLP) | Deep learning MLP that uses a combination of input, hidden, and output layers using a large number of neurons for building the relationship between the predictors | caret (R); Keras (R and Python); Scikit-learn (Python) | [23] |

## 5. Genotypic Data and Major Genes

The development of next-generation sequencing (NGS) allowed an exponential advance in genotyping driven by the goal of sequencing different genomes. Sequencing has improved with the implementation of parallel sequencing that allowed polymorphism discovery, gene expression analysis, and population genotyping. The cost of genotyping has allowed the application of NGS and revolutionized applied plant breeding [106]. Before NGS, MAS was the primary use of genotypic data in selection. However, the advantages of GS over MAS have been observed in many studies [21,32,36,107]. Markers for major genes used in MAS in breeding programs for traits such as disease resistance can still be utilized with the integration of MAS and GS. Furthermore, markers for major genes and significant loci derived from GWAS can be integrated into GS models such as rrBLUP as fixed effects and help account for genetic variance and selection accuracy [20,21,32,108]. The advantage of integrating the major markers varies. For example, Rutkoski et al. [20] showed a significant increase in prediction accuracy for quantitative stem rust (*Puccinia graminis* f. sp. *tritici*). Conversely, integrating major markers for stripe rust (*Puccinia striiformis* f. sp. *tritici*) has been demonstrated to have little or no increase in prediction accuracy [32].

Significant markers from GWAS can also be integrated into GS models; however, negligible increases in prediction accuracy have been found. Publicly available GWAS markers were integrated into GS models, but accuracy only increased by 1% [109]. Therefore, population-specific *de novo* GWAS markers were integrated. Arruda et al. [21] demonstrated an increase in accuracy of 14% when integrating significant *de novo* GWAS markers for Fusarium head blight (*Fusarium graminearum* Schwabe) and Spindel et al. [110] demonstrated an increase in accuracy by 10%. In contrast, Merrick et al. [32] and Rice and Lipka [108] demonstrated a decrease in prediction accuracy by using *de novo* GWAS markers for simulated traits across various types of genetic architectures. The lack of increase in accuracy can be a consequence of the GS models already accounting for the majority of variation of the trait in the genome-wide markers or that the major marker may not account for enough phenotypic variation [32]. Therefore, major genes should be integrated into GS models on a population basis; and further, as Bernardo [111] proposed, only markers that account for more than 10% of the variation should be incorporated into GS models.

## 6. Real World Applications

The first practical application of GS for selecting lines in small grains was published by Asoro et al. [11] in oats. β-glucan was selected and compared by GS, MAS, and PS. In this study, GS and MAS increased β-glucan in their resulting populations. GS has also been applied on large scale at CIMMYT, Mexico, since 2010 [112]. CIMMYT has explored the optimization of various aspects of GS, where it is currently implemented to increase the accuracy of line selection in the PD portion of the CIMMYT spring wheat program. At CIMMYT, GS is implemented to select lines in the same generation or to select lines in earlier generations with two selection cycles annually [112]. As the training population grew, newer genotyping technology and improved GS models were implemented, consequently improving the GS accuracy for grain yield within the CIMMYT breeding program over the last ten years. In the case of grain yield, for example, CIMMYT was able to predict low-performing lines for culling; however, finding the top 10% performing lines imposed some difficulty [112]. Additionally, the prediction accuracy for disease resistance and end-use quality traits were high, with values reaching up to 83% [112]. Therefore, GS has the ability to discard lines for grain yield but should be used with caution when selecting top-performing lines in the PD pipeline. However, there was a significant reduction in costs for the implementation of GS in early generation yield trials with low replication for grain yield and disease resistance, and no testing for end-use quality was required [112].

A new trend in public breeding programs is to leverage the resources of multiple breeding programs to efficiently phenotype early-stage lines and integrate GS consortiums (GSC). These GSCs increase the size of programs and screening environments without increasing the investment and resource allocation in a single program [113]. Lines are phenotyped in some or all programs in sparse testing schemes, and the GS is implemented to predict local (single program) and broad (multiple programs) values of lines. In addition, the GSC allows a common genotyping platform to implement large-scale genotyping to increase the size of the training populations without increasing investments. Further, large-scale GS allows the prediction of the performance of lines or traits not phenotyped in all environments. However, Sneller et al. [113] indicated the importance of creating training populations with related germplasm from each program rather than utilizing all lines. Overall, GSCs have the ability to increase the size of individual breeding programs and the accuracy of GS methods without increasing resource allocation and investments [113].

## 7. Conclusions

In our review, we explored optimizing a breeding program for GS based on aspects of the breeder's equation. We outlined the need to redesign the PD pipeline from the ground up by integrating speed breeding and double-haploid technologies and implementing newer field designs while optimizing training populations in an effort to increase statistical power to increase selection accuracy and genetic gains. By utilizing GS and leveraging the existing program's phenotypic data as well as the multi-environment trials, the PD pipeline can be optimized to increase selection accuracy and genetic gain. Multi-environment models can account for GE for complex traits, whereas multi-trait models can take advantage of the genetic correlation of highly heritable traits to increase the prediction accuracy of complex and low heritable traits. In addition, newer methodologies to integrate environmental variables and HTP can aid GS models as well as utilize newer statistical models such as DL to improve selection accuracy. Therefore, by redesigning the breeding program to take advantage of the plethora of new technologies while optimizing components based on the breeder's equation, we can change the traditional thinking of breeding as a "numbers game" to a more precise and efficient "chess game" to maximize resources and exponentially increase genetic gains and improve new varieties.

**Author Contributions:** Conceptualization, L.F.M.; writing—original draft preparation, L.F.M., A.W.H., and K.S.S.; writing—review and editing, L.F.M., A.W.H., K.S.S., D.N.L. and A.H.C.; supervision, A.H.C.; funding acquisition, A.H.C.; All authors have read and agreed to the published version of the manuscript.

**Funding:** This research was partially funded by the National Institute of Food and Agriculture (NIFA) of the U.S. Department of Agriculture (Award number 2016-68004-24770 and 2022-68013-36439), Hatch project 1014919, and the O.A. Vogel Research Foundation at Washington State University.

**Institutional Review Board Statement:** Not applicable.

**Informed Consent Statement:** Not applicable.

**Data Availability Statement:** Not applicable.

**Conflicts of Interest:** The authors declare no conflict of interest.

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
