# Peer review of "Optimizing Plant Breeding Programs for Genomic Selection"

_agronomy, doi:10.3390/agronomy12030714_

Round 1

Reviewer 1 Report

The present review on Optimizing plant breeding programs for genomic selection is well narrated and included most of the potential methods/techniques to improve the genetic selection for different traits in crop breeding programs, particularly in wheat breeding programmes. Most of the methods mentioned in this review are being used in current crop breeding programs but use of speed breeding and double haploid will definitely fasten the development process. Among multi environmental models, two step GBLUP is more effective in geneomic selection. 

Need to elaborate how to tackle the spatial variation in genomic selection process. 

It would be great to write deep learning/machine learning section in more simple words. 

Author Response

Reviewer 1:

  1. The present review on Optimizing plant breeding programs for genomic selection is well narrated and included most of the potential methods/techniques to improve the genetic selection for different traits in crop breeding programs, particularly in wheat breeding programmes. Most of the methods mentioned in this review are being used in current crop breeding programs but use of speed breeding and double haploid will definitely fasten the development process. Among multi environmental models, two step GBLUP is more effective in geneomic selection. 

Thank you for taking the time to review our manuscript, and we hope our revisions have addressed your concerns.

  1. Need to elaborate how to tackle the spatial variation in genomic selection process. 

Thank you for your comments. We discussed the use of field design and spatial corrections and their effects on genomic selection accuracy in the two paragraphs from lines 249-268. We added some clarifications so it’s clear we talk about the effect on prediction accuracy.

  1. It would be great to write deep learning/machine learning section in more simple words. 

Thank you for your comments. We have tried to clarify in the paragraph from 424-449 to allow better understanding of DL. However, DL models are complex and harder to understand, and we would need a much larger section to explain the intricacies adequately.

Reviewer 2 Report

The manuscript comprehensively reviewed the various methods for optimizing genomic selection in plant breeding program. The figure is clearly presented, and conclusions are supported by multiple evidence. The review is particularly interesting for plant geneticists and breeders, and it is OK.

Author Response

Reviewer 2:

  1. The manuscript comprehensively reviewed the various methods for optimizing genomic selection in plant breeding program. The figure is clearly presented, and conclusions are supported by multiple evidence. The review is particularly interesting for plant geneticists and breeders, and it is OK.

Thank you for taking the time to review our manuscript.

Reviewer 3 Report

Dear Authors,

the manuscript tackles major topics of Genomic Selection which are important from practical/application point of view for product development/wheat variety breeding. I appreciated this manuscript very much, there are many very interesting citations, unfortunately, I cannot judge on the GS models itself, this is not my area of expertise, I am a wheat breeder utilizing GS for severals years. I feel that this paper can be a good summary for new PhD students or young breeders to start with GS.

The manuscript is well structured and generally clear to me. I cannot too much judge on the completeness of the references, sorry.

Anyhow, I very much appreciated the statements and conclusions made! It really tackles many of my thoughts and questions from practical point of view and this review is therefore very valuable. I am not aware whether anything similar has been recently published.

There arweonly two small questions from my side: 

line 443+444 - I do not understand.... displayed higher acuracy in two of three data sets but had similar accuracy.... - probably you meant something different than what you stated?

lines 491-495. I wonder - I guess, you meant 14% (Arruda et al.), not 0,14%!!! 14% seems right when looking up the Arruda paper! What about 0,01% in line 491 and 0,10% in line 495 - probably these should be 1% and 10% - otherwise it does not make much sense to me (I did not look up the papers No. 107 and 108 - I only checked paper No. 22.

Good luck for your further work!

Your reviewer

Author Response

Reviewer 3:

  1. the manuscript tackles major topics of Genomic Selection which are important from practical/application point of view for product development/wheat variety breeding. I appreciated this manuscript very much, there are many very interesting citations, unfortunately, I cannot judge on the GS models itself, this is not my area of expertise, I am a wheat breeder utilizing GS for severals years. I feel that this paper can be a good summary for new PhD students or young breeders to start with GS.

The manuscript is well structured and generally clear to me. I cannot too much judge on the completeness of the references, sorry.

Anyhow, I very much appreciated the statements and conclusions made! It really tackles many of my thoughts and questions from practical point of view and this review is therefore very valuable. I am not aware whether anything similar has been recently published.

Thank you for taking the time to review our manuscript

There arweonly two small questions from my side: 

  1. line 443+444 - I do not understand.... displayed higher acuracy in two of three data sets but had similar accuracy.... - probably you meant something different than what you stated?

You are correct and thank you for pointing that out. We have removed the rest of the sentence, and the new sentence is on line 464.

  1. lines 491-495. I wonder - I guess, you meant 14% (Arruda et al.), not 0,14%!!! 14% seems right when looking up the Arruda paper! What about 0,01% in line 491 and 0,10% in line 495 - probably these should be 1% and 10% - otherwise it does not make much sense to me (I did not look up the papers No. 107 and 108 - I only checked paper No. 22.

Thank you for the comment. We change the numbers to percentages like you suggested.